# Serum Ceramides as Prognostic Biomarkers of Large Thrombus Burden in Patients with STEMI: A Micro-Computed Tomography Study

**DOI:** 10.3390/jpm11020089

**Published:** 2021-01-31

**Authors:** Efstratios Karagiannidis, Andreas S. Papazoglou, Nikolaos Stalikas, Olga Deda, Eleftherios Panteris, Olga Begou, Georgios Sofidis, Dimitrios V. Moysidis, Anastasios Kartas, Evangelia Chatzinikolaou, Kleoniki Keklikoglou, Andreana Bompoti, Helen Gika, Georgios Theodoridis, Georgios Sianos

**Affiliations:** 1First Department of Cardiology, AHEPA University Hospital, Aristotle University of Thessaloniki, St. Kiriakidi 1, 54636 Thessaloniki, Greece; stratoskarag@gmail.com (E.K.); anpapazoglou@yahoo.com (A.S.P.); nstalik@gmail.com (N.S.); g_sofidis@yahoo.gr (G.S.); dimoysidis@gmail.com (D.V.M.); tkartas@gmail.com (A.K.); 2Laboratory of Forensic Medicine and Toxicology, School of Medicine, Aristotle University of Thessaloniki, 54124 Thessaloniki, Greece; oliadmy@gmail.com (O.D.); eleftherios.panteris@gmail.com (E.P.); gkikae@auth.gr (H.G.); 3Biomic_AUTh, Center for Interdisciplinary Research and Innovation (CIRI-AUTH), Balkan Center, B1.4, 10th km Thessaloniki-Thermi Rd, P.O. Box 8318, 57001 Thessaloniki, Greece; olina_18@hotmail.com (O.B.); gtheodor@chem.auth.gr (G.T.); 4Laboratory of Analytical Chemistry, Department of Chemistry, Aristotle University of Thessaloniki, 54636 Thessaloniki, Greece; 5Institute of Marine Biology, Biotechnology and Aquaculture (IMBBC), Hellenic Centre for Marine Research (HCMR), 71500 Heraklion, Crete, Greece; evachatz@hcmr.gr (E.C.); keklikoglou@hcmr.gr (K.K.); 6Biology Department, University of Crete, 71003 Heraklion, Crete, Greece; 7Medway NHS Foundation Trust, Gillingham ME7 5NY, UK; andreana.bompoti@gmail.com

**Keywords:** ST-elevation myocardial infarction, thrombus, micro-CT, thrombus aspiration, ceramides

## Abstract

ST-elevation myocardial infarction (STEMI) remains one of the leading causes of mortality worldwide. The identification of novel metabolic and imaging biomarkers could unveil key pathophysiological mechanisms at the molecular level and promote personalized care in patients with acute coronary syndromes. We studied 38 patients with STEMI who underwent primary percutaneous coronary intervention and thrombus aspiration. We sought to correlate serum ceramide levels with micro-CT quantified aspirated thrombus volume and relevant angiographic outcomes, including modified TIMI thrombus grade and pre- or post-procedural TIMI flow. Higher ceramide C16:0 levels were significantly but weakly correlated with larger aspirated thrombus volume (Spearman r = 0.326, *p* = 0.046), larger intracoronary thrombus burden (TB; *p* = 0.030) and worse pre- and post-procedural TIMI flow (*p* = 0.049 and *p* = 0.039, respectively). Ceramides C24:0 and C24:1 were also significantly associated with larger intracoronary TB (*p* = 0.008 and *p* = 0.001, respectively). Receiver operating characteristic analysis demonstrated that ceramides C24:0 and C24:1 could significantly predict higher intracoronary TB (area under the curve: 0.788, 95% CI: 0.629–0.946 and 0.846, 95% CI: 0.706–0.985, respectively). In conclusion, serum ceramide levels were higher among patients with larger intracoronary and aspirated TB. This suggests that quantification of serum ceramides might improve risk-stratification of patients with STEMI and facilitate an individualized approach in clinical practice.

## 1. Introduction

Large thrombus burden (TB) in patients with ST-elevation myocardial infarction (STEMI) constitutes an independent risk factor for mortality and for adverse clinical and angiographic outcomes, including distal embolization, no-reflow phenomenon and stent thrombosis [1,2,3,4]. Although the routine use of manual aspiration thrombectomy (MATh) is not recommended in patients with STEMI according to the most recent European Society of Cardiology (ESC) guidelines, patients with large pre-procedural TB could benefit from MATh [5,6].

The integration of novel biomarkers, derived from patients’ metabolomic profiling, could provide complementary prognostic information, thereby improving risk-stratified patient management [7]. Ceramides constitute members of the sphingolipid family, which support the structure of the membrane of eukaryotic cells and mediate multiple cell-signaling pathways. Emerging evidence suggests that aberrant accumulation of ceramides has been linked to the development and progression of atherosclerosis by promoting low density lipoprotein infiltration to the endothelium [8,9]. Recent data also support their role as determinants of plaque components and predictors of plaque rupture in patients with STEMI [10], rendering these bioactive sphingolipids useful indicators for STEMI risk-stratification.

In parallel, advances in cardiovascular imaging facilitate the quantification of characteristics, which have been subjective to date [11,12,13,14]. Identification of novel imaging parameters enables patient-specific predictions of adverse outcomes [15]. Current research has demonstrated the potential for using micro-CT to quantitatively and qualitatively assess extracted thrombotic material characteristics in STEMI.

In this paper we used data from micro-CT scans of extracted thrombi from patients with STEMI. We correlated the derived volumetric findings with levels of serum ceramides, aiming to open the door for a novel personalized approach in patients with STEMI.

## 2. Materials and Methods

The QUEST-STEMI study enrolled 113 STEMI patients to assess aspirated thrombus burden characteristics with micro-CT and to explore potential associations with adverse angiographic and electrocardiographic outcomes [16]. A subset of 38 patients, reporting at least 8 h fasting, was simultaneously enrolled in the CorLipid trial, which aims to evaluate the diagnostic utility of patients’ metabolic signature for the determination of the severity of coronary artery disease [17].

The present analysis examined patients who were co-enrolled in both studies. Detailed inclusion and exclusion criteria of this study are presented in Table 1. We sought to explore the correlation of serum ceramide levels with aspirated thrombus volume (as quantified via micro-CT Sky-Scan 1172(Bruker, Kontich, Belgium)), as well as with angiographic outcomes in STEMI patients undergoing primary percutaneous coronary intervention (PCI) and MATh. For enhanced micro-CT analysis of aspirated clots (Figure 1), phosphotungstic acid was used as a contrast agent for their staining. The detailed protocol of micro-CT scanning has been previously described [16].

Venous blood sample was collected from all participants prior to coronary angiography. Ceramide species C16:0, C18:0, C24:0 and C24:1, namely: N-Palmitoyl-D-erythro-sphingosine (Cer d18:1/16:0), N-stearoyl-D-erythro-sphingosine (Cer d18:1/18:0), N-lignoceroyl-D-erythro-sphingosine (Cer d18:1/24:0) and N-nervonoyl-D-erythro-sphingosine (Cer d18:1/24:1), were quantified via Ultra High Pressure Liquid Chromatography- tandem Mass Spectrometry (UHPLC-MS/MS). Outcomes assessed were the association of ceramides levels with: (1) aspirated thrombus volume (as quantified by micro-CT) divided by Reference Vessel Diameter (volume/RVD), (2) angiographic modified TIMI thrombus grade classification [18], (3) pre-procedural and post-procedural TIMI flow and (4) biomarkers of thrombosis (d-dimers) and myocardial necrosis (high-sensitivity troponin, CK-MB).

The association of serum ceramide levels with outcomes was assessed with use of Spearman’s correlation and logistic regression. Data analysis was executed via SPSS version 26.0 (SPSS software, Chicago, IL, USA) software and *p* values < 0.05 were considered statistically significant.

## 3. Results

The baseline clinical and demographic characteristics of the patients included in our study are presented in Table 2. Our results (Table 3, Figure 2) indicate a significant, but weak positive Spearman’s correlation between ceramide C16:0 levels with volume/RVD values (*p* = 0.046, r = 0.326), and a non-significant trend towards the association of C18:0 and C24:1 levels with volume/RVD values (*p* = 0.069, r = 0.299 and *p* = 0.059, r = 0.309, respectively). Serum C16:0 (*p* = 0.030, Nagelkerke R^2^ = 0.236), C24:0 (*p* = 0.008, Nagelkerke R^2^ = 0.311) and C24:1 levels (*p* = 0.001, Nagelkerke R^2^ = 0.423) were also higher among patients with higher modified TIMI thrombus grade. Moreover, elevated C16:0 levels were significantly associated with worse pre-procedural TIMI flow (*p* = 0.049, Nagelkerke R^2^ = 0.210). Finally, elevation in ceramide C16:0 and C18:0 levels was significantly linked with worse post-procedural TIMI flow (*p* values= 0.039 and 0.017, respectively).

Additionally, we performed receiver operating characteristic (ROC) analysis to investigate the sensitivity and specificity of ceramides as predictive parameters of intracoronary thrombus burden and propose exact prognostic cut-off values of ceramide levels. The ROC curves, depicted in Figure 3, demonstrate that higher serum levels of ceramides 24:0 and 24:1 were significantly associated with higher intracoronary TB, as assessed via modified TIMI classification (area under the curve: 0.788, 95% CI: 0.629–0.946 and 0.846, 95% CI: 0.706–0.985, respectively). Specifically, ceramide C24:0 levels above 6.77 μΜ and C24:1 above 2.71 μΜ were significant predictive biomarkers of higher thrombus grade with sensitivity of 80% & 85% and specificity 66% & 75%, respectively. Regarding ceramide C16:0 levels, a non-significant predictive value was observed with area under the curve: 0.698, 95% CI: 0.493–0.903. Lastly, the Spearman correlation analysis between ceramide levels and clinically useful biomarkers (high-sensitivity troponin, CK-MB and d-dimers) did not yield any significant outcome (*p*-values > 0.05).

## 4. Discussion

Our findings suggest that serum ceramide levels are higher among patients with larger intracoronary and aspirated TB, although some correlations observed were weak. Elevated C16:0 levels were also found in patients with worse pre- and post-procedural TIMI flow. Hence, ceramides could be potential biomarkers of high thrombotic state in STEMI patients. Consequently, they could be used as predictive factors for the development of risk-stratification models in STEMI, with potential for evolving into a novel tool for personalized medicine.

The need for patient-level risk stratification in patients presenting with STEMI is reflected in the results of large randomized clinical trials and meta-analyses, concluding that MATh may be considered only in certain patients [19,20,21]. However, the recent guidelines have not provided specific evidence about the profile of patients, in whom MATh should be undertaken and performing MATh is left at the discretion of the interventional cardiologist [22]. Presently, emerging evidence shows that patients with large pre-procedural TB could benefit from MATh and, therefore, ceramides could be employed as a part of a more sophisticated risk stratification algorithm, which can accurately identify patients with STEMI with potential benefit from MATh [6,23,24]. Of note, the total run time of UPLC-MS/MS is within the limits suggested by the current ESC guidelines on the management of patients with STEMI (maximum time from STEMI diagnosis to wire crossing: 60 min in primary PCI hospitals). Hence, ceramides quantification could be truly timely and feasible for the prediction of thrombotic burden in a patient with STEMI, provided that it does not delay reperfusion.

Our results are consistent with previous studies which demonstrated significant association between elevated ceramide levels and increased coronary atherosclerotic burden in STEMI patients [9,25]. Besides the proatherogenic role that ceramides may exert, studies have also documented that distinct ceramides are associated with specific plaque characteristics (plaque rupture, higher necrotic core fraction or higher lipid core burden) or increased cardiovascular and cerebrovascular risk [26,27,28]. However, our study was not designed to assess hard clinical endpoints and our findings should also be interpreted taking into consideration the single-center nature of the study, the restricted sample of participants and the absence of a comparative control group of patients with non-ST-elevation acute coronary syndrome.

In conclusion, quantification of serum ceramides might improve risk-stratification of patients with STEMI and guide future decision-making in a more individualized approach. Further research is warranted to explore the association of TB with ceramide levels and elucidate whether these sphingolipid products could be employed as potential diagnostic or therapeutic targets.

## Figures and Tables

**Figure 1 jpm-11-00089-f001:**
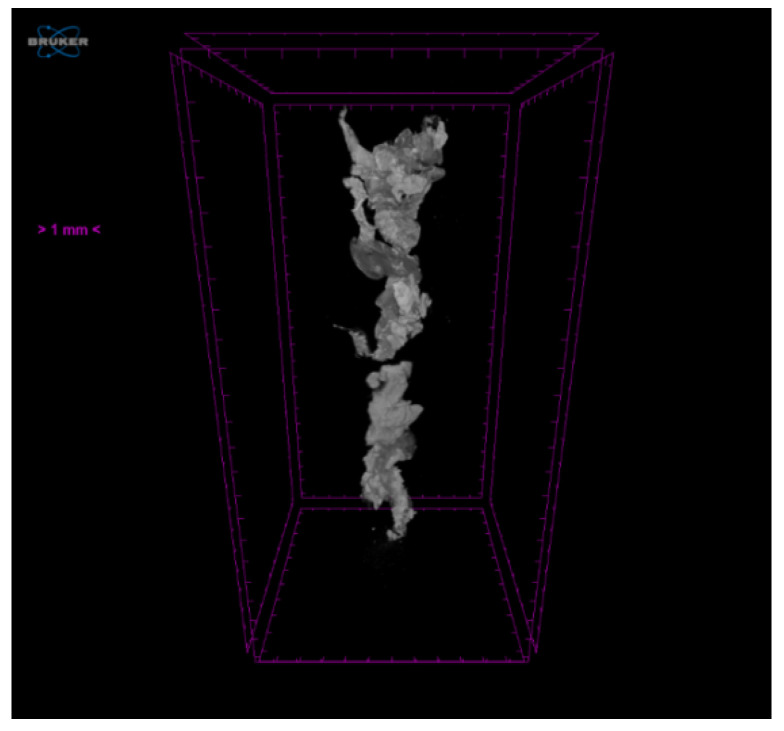
3-D volume rendering of a thrombus using the Skyscan 1172 micro-CT scanner at Hellenic Centre for Marine Research. The clot was stained using 0.3% phosphotungstic acid as a contrast agent. Projection images were reconstructed into sections (cross-section images) via NRecon (Bruker, Kontich, Belgium) software.

**Figure 2 jpm-11-00089-f002:**
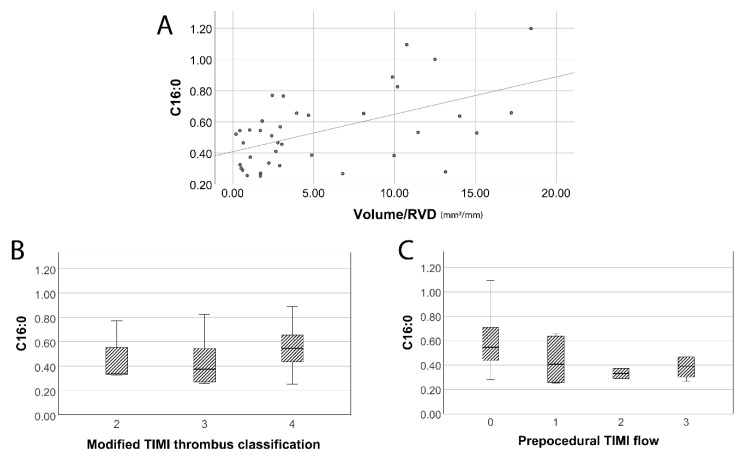
(**A**) Serum ceramide C16:0 levels were positively correlated with larger aspirated thrombus volume. (**B**) The median C16:0 level of patients with TIMI thrombus Grade 4 was higher compared to the corresponding levels of patients with lower TIMI thrombus Grades (2 and 3). (**C**) The median C16:0 levels were higher in patients with worse pre-procedural TIMI flow.

**Figure 3 jpm-11-00089-f003:**
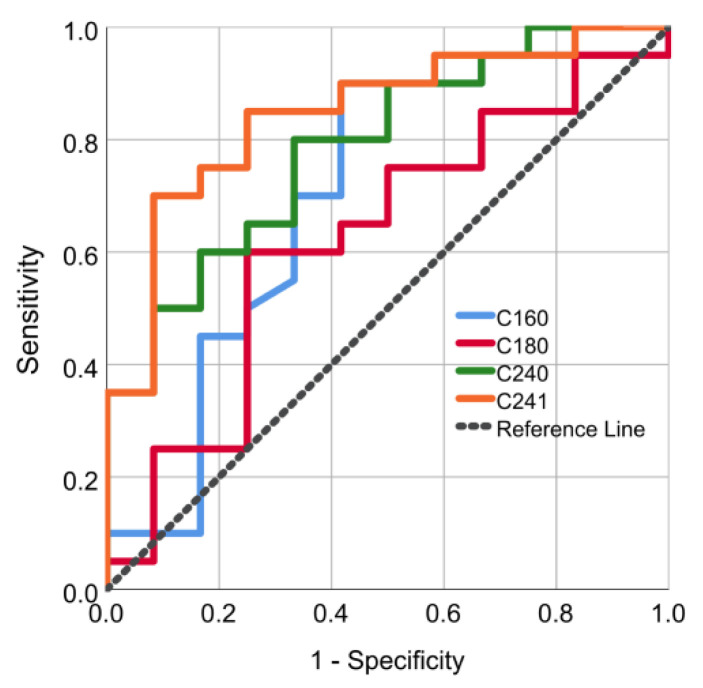
ROC curve analysis. Receiver-operating characteristic curve analysis of thrombus grade diagnosed by ceramide C16:0, C18:0, C24:0 and C24:1 levels.

**Table 1 jpm-11-00089-t001:** Inclusion and exclusion criteria of the study.

Inclusion Criteria:	Exclusion Criteria:
1. STEMI diagnosis	1. Patients receiving fibrinolytic therapy
2. Patients undergoing both primary PCI and MATh within 12 h from symptom onset	2. Patients with known intolerance to heparin or anti-platelet medication
3. Fasting plasma for at least 8 h	

**Table 2 jpm-11-00089-t002:** (**A**) Baseline clinical and demographic patient characteristics and (**B**) outcomes of interest.

**A. Baseline Clinical and Demographic Patient Characteristics**	
Age [years(± SD)]	58.4 ± 12.5
Male gender [n (%)]	30 (78.9%)
Smoking [n (%)]	27 (71.1%)
Hypertension [n (%)]	13 (34.2%)
Dyslipidemia [n (%)]	7 (18.4%)
Diabetes mellitus [n (%)]	3 (7.9%)
BMI [kg/m^2^ (± SD)]	32.1 ± 23.2
Pain-to-balloon time [min (± SD)]	305 ± 250.5
Prior Medication [n (%)]:	
I. aspirin	5 (13.8%)
II. clopidogrel	2 (5.3%)
III. statin	8 (21.1%)
anticoagulant(s)	2 (5.3%)
Peri-procedural Medication [n (%)]:	
I. heparin	1 (2.6%)
II. clopidogrel	7 (18.4%)
III. prasugrel	1 (2.6%)
IV. ticagrelor	32 (84.2%)
GP2B/3A	17 (44.7%)
**B. Outcomes of Interest**	
Aspirated thrombus volume [m^3^ (± SD)] Pre-procedural TIMI flow	16.6 ± 18.3
0	24 (63.2%)
I	6 (15.8%)
II	2 (5.3%)
III	6 (15.8%)
Post-procedural TIMI flow	
0	1 (2.6%)
I	1 (2.6%)
II	5 (13.2%)
III	31 (81.6%)
Modified TIMI thrombus grade classification	
Grade 2	6 (15.8%)
Grade 3	10 (26.3%)
Grade 4	22 (57.9%)

**Table 3 jpm-11-00089-t003:** Association of serum ceramides levels with study outcomes.

	Volume/RVD	TIMI Thrombus Classification	Pre-Procedural TIMI Flow	Post-Procedural TIMI Flow
C16:0	**r = 0.326** ***p* = 0.046**	**Nagelkerke *R*^2^ = 0.236** ***p* = 0.030**	**Nagelkerke *R*^2^ = 0.210** ***p* = 0.049**	**Nagelkerke *R*^2^ = 0.277** ***p* = 0.039**
C18:0	r = 0.299 *p* = 0.069	Nagelkerke *R*^2^ = 0.119 *p* = 0.188	Nagelkerke *R*^2^ = 0.129 *p* = 0.208	**Nagelkerke *R*^2^= 0.329** ***p* = 0.017**
C24:0	r = 0.214 *p* = 0.197	**Nagelkerke *R*^2^ = 0.311** ***p* = 0.008**	Nagelkerke *R*^2^ = 0.055 *p* = 0.601	Nagelkerke *R*^2^ = 0.218 *p* = 0.092
C24:1	r = 0.309 *p* = 0.059	**Nagelkerke *R*^2^ = 0.423** ***p* = 0.001**	Nagelkerke *R*^2^ = 0.177 *p* = 0.095	Nagelkerke *R*^2^ = 0.075 *p* = 0.556

Statistically significant findings (*p*-values < 0.05) are marked in bold.

## Data Availability

Data are available from G.S. (Georgios Sianos) (e-mail: gsianos@auth.gr) upon reasonable request and with permission of AHEPA University Hospital.

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
