# Peer review of "Serum Ceramides as Prognostic Biomarkers of Large Thrombus Burden in Patients with STEMI: A Micro-Computed Tomography Study"

_jpm, 2021, doi:10.3390/jpm11020089_

Round 1

Reviewer 1 Report

The authors explored the association between circulating ceramides and STEMI-related angiographic outcomes and an intervention that estimates thrombotic burden. The latter is highly interesting to elucidate bioactive lipids, such as cermides, link with thrombotic burden.  However, a major limitation to this study is low power combined with weak correlations, which is also highlighted by the authors. Would these correlation be the same for non-STEMI patients with the same TB burden?? A control group would be most appreciated if possible!

Since circulating lipids are highly confounded by diet, my main question is; were they fated at the sampling occasion? Please clarify!

The authors denote the cermides acyl-group but no information of the backbone acyl composition (d18:1 or d18:0). Have the authors data on this? If so must be provided. If not please discuss and refer to the literature of the most likely backbone composition.

Text errors.

References numbers in text are missing brackets.

Cermides have shown to increase with LDL cholesterol

Axis values on figure 2 is too small.

Not needed hyphenations are found in places throughout the text words. Please revise!

Page 2, line 76, please define PCI.

Reviewer 2 Report

The study by Karagiannidis et al aims to demonstrate a link between Micro-CT measured thrombus volume, TIMI flow and ceramide levels in 38 STEMI patients. The authors claim that ceramide levels and in particular C16:0 predicts thrombus volume and TIMI flow and may be used as biomarker for a prothrombotic state. However, even though the authors claim that their study „aims to open the door for a novel personalized approach“ they fail to demonstrate sufficient evidence to substantiate their findings and merely demonstrate associations of ceramide levels and thrombus volume and TIMI flow respectively. At least an additional C-statistic should have been performed to address the diagnostic value of ceramides. 

In addition, I have a few remarks regarding the MS:

  • It remains elusive what kind of patients were included. Please add patient characteristics to the MS (in particular medication that may have affecte thrombus formation – P2Y12, a2bB3, Heparin etc). Also: In and exclusion criteria should be mentioned.
  • Please briefly state the exclusion criteria or was every patient included in a consecutive manner?
  • Is there any correlation of ceramide levels with other markers of thrombosis -> e.g. d-dimers, ROTEM, fibrinogen levels?
  • Is there any relation with ceramide levels and severity of MI? E.g. do patients with Myocardial Infarciton have more severe infarciton  higher CK, Troponin or CK-MB?
  • Is Ultra High Liquid Chromatography – tandem Mass Spectrometry feasible (i.e. timely?) to predict pro-thrombotic burden in a STEMI patient?

Minor:

The MS suffers plenty formation errors which should be corrected:

  • There are many unnecessary conjunctions present in the MS which should be removed (E.g. spohisticat-ed; be-sides; be-tween)
  • Please use super-scripted citations. Also the current bibliography appears incorrect and should be changed according to the MDPI recommendations.

Round 2

Reviewer 2 Report

I would like to thank the authors for their revision. I have no further comments.